# A Study on Defect Detection of Dissimilar Joints in Cu-STS Tubes Using Infrared Thermal Imaging of Induction Heating Brazing

Chung-Woo Lee [1,2] 🆔, Suseong Woo [1] and Jisun Kim [1,*]

1 Automotive Materials & Components R&D Group, Korea Institute of Industrial Technology, Gwangju 61012, Republic of Korea; leecw@kitech.re.kr (C.-W.L.); dfsin123@kitech.re.kr (S.W.)
2 Department of Metallurgical Engineering, Jeonbuk National University, Baekje-daero, Deokjin-gu, Jeonju-si 54896, Republic of Korea
* Correspondence: kimjisun@kitech.re.kr; Tel.: +82-62-600-6298

**Abstract:** We proposed a novel detection method for identifying joint defects in the brazing process between copper tubes and stainless steel using a convolutional neural network (CNN) model. The brazing joints were created using high-frequency induction heating equipment, and infrared thermal imaging cameras were employed to capture the thermal data generated during the jointing process. The experiments involved 15.88 mm diameter copper tubes commonly used in plate heat exchangers, stainless-steel tubes, and filler metal containing 20% Ag. The thermal data were obtained with a resolution of $80 \times 80$ pixels per frame, resulting in 4796 normal joint data and 5437 defective joint data collected over 100 high-frequency induction-heating brazing experiments. A total of 10,233 thermal imaging data were categorized into 6548 training data, 1638 validation data, and 2047 test data for the development of the predictive model. We designed CNN models with varying hyperparameters, specifically the number of kernel filters and nodes, to evaluate their impact on detection performance. A comparative analysis revealed that a CNN model structure, exhibiting 98.53% accuracy and 99.82% recall on test data, was the most effective. The selected CNN-based defect prediction model demonstrated the potential of using CNN models to discern joint defects in tube configurations that are challenging to identify visually. This study opens avenues for applying CNN-based models for detecting imperfections in complex tube structures.

**Keywords:** brazing; high-frequency Induction heating; defect identification; infrared thermal image; convolutional neural network

## 1. Introduction

The continued consumption of fossil fuel poses a serious problem in the fields of energy and the environment. Modern society demands more measures for environmental protection, and some of these issues can be addressed by reducing energy dependence and increasing energy efficiency. In the industrial sector, one solution to enhance energy efficiency and decrease dependence on fossil fuels is to increase the efficiency of heat exchange [1,2]. Plate heat exchangers, known for their wide heat-exchange area, as compared to other heat exchangers, are compact and lightweight. They facilitate easy expansion of heat-exchange capacity, making them widely used in various heat-exchange applications [3–5]. Plate heat exchangers typically join copper tubes to bodies made of stainless steel. Although the joints between the copper tubes and the stainless-steel bodies are typically brazed, inspections of the joint quality are essential due to potential defects, such as cracks and insufficient penetration. These defects can lead to a rapid decline in the heat efficiency of the heat exchanger [6–8]. Generally, the joint region between the copper tube and the stainless-steel materials is situated internally, making it challenging to visually inspect for defects. An assessment of joint quality typically relies on water immersion and

non-destructive testing. However, water immersion requires time for drying the wet tubes post-inspection, and non-destructive testing involves expensive equipment and specialized personnel, resulting in significant costs and time consumption. Therefore, there is an urgent need for research in defect detection that can save costs and time in order to enhance product quality and productivity [9].

Recent research in the field of welding and joining has observed the continuous use of algorithms, such as those used in machine learning, to predict defects and characteristics. Mishra et al. [10] conducted a characteristic prediction study using machine learning to predict the maximum tensile strength of joints in resistance spot welding. Chen et al. [11] predicted joint quality in resistance spot welding using a machine-learning parallel-strategy algorithm based on the processing of feature data from the welding process. Kaiser et al. [12] employed deep neural networks (DNN) and random forests (RF) to predict damage modes and failure strength in single-lap joints (SLJs). Elsheikh et al. [13] provided a comprehensive review of the application of various machine-learning methods in friction stir welding (FSW) and demonstrated superior performance in machine-learning techniques, as compared to traditional statistical methods. While previous studies have utilized machine learning to predict joint quality in various joining methods, the research into predicting internal defects is limited. Perri et al. [14] presented a CNN model named WelDeNet that was based on images obtained by transmitting radiation to the welding bead and showed it could classify 99.5% of welding defects. Munir et al. [15] compared the prediction performance of DNN and CNN models on the noisy ultrasonic signatures found in detecting welding defects through ultrasonic testing, confirming that the CNN models performed better, even with noisy signals. Through several previous studies, it was recognized that technologies like machine learning offered potential benefits in accurately classifying defects while reducing inspection costs. It was established that machine-learning models, such as CNNs, could be applied to predict internal defects [16–19]. Furthermore, utilizing predictive models in the process was recognized for its quantitative reduction in the time required, as well as improved reliability in accuracy, as compared to conventional manual fault-classification methods, as evidenced by numerous studies [20–23]. However, a fundamental challenge has remained as the data used in the prediction models were based on data obtained through non-destructive testing, which presented a significant inherent cost issue.

To address these issues, there is a need for methods that can predict internal defects at a lower cost. In this study, we proposed a method for predicting joint defects in brazing joints, which are predominantly applied to small components and have had relatively few studies conducted, as compared to various welding and joining methods. While research has been conducted on predicting welding defects using CNN-based predictive models, there has been no research on predicting joint defects using thermal imaging data acquired during the brazing of tube-shaped joints. Therefore, to predict defects in joints, we utilized real-time data acquired through infrared thermal imaging cameras during the brazing joint process between copper tubes and stainless-steel bodies using high-frequency induction heating. Thermal imaging data measured the temperature changes across the entire joint, and the presence of defective joints was determined by examining a cut section of the completed specimen. To develop a high-performance predictive model, we compared the predictive performance based on hyperparameters and derived a CNN model that exhibited the optimal performance on the test data. The results of this study are expected to lay the foundation for low-cost, real-time detection of brazing joint defects in practical applications.

## 2. Experimental Method and Design

### 2.1. Experimental Setup

To facilitate a brazing joint between copper and stainless-steel tubes, a high-frequency induction heating system was configured using a power source (Osung hitech (Daegu, Republic of Korea), 75~100 KW), chiller (Osung hitech (Republic of Korea)), and jig system. Infrared thermal imaging data used for training the predictive model were acquired through an infrared thermal camera (Micro epsilon (Sant'Antonio, Italy), Thermo IMAGERTIM-

40) and a self-developed monitoring system. Figure 1 illustrates the configuration of the high-frequency induction heating brazing joint system used in this study.

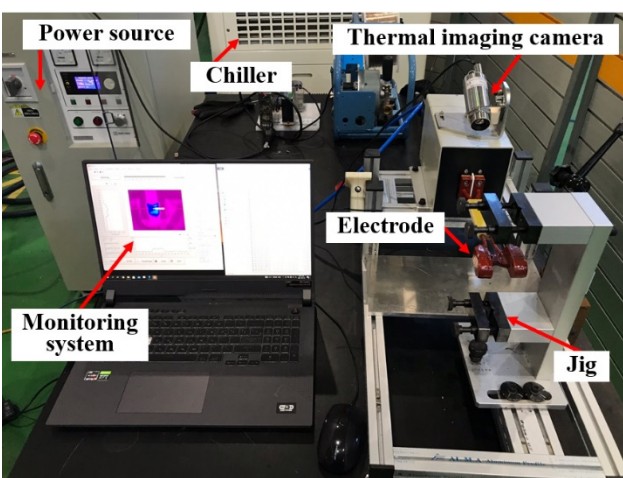

**Figure 1.** Experimental setup for brazing joint system.

The copper tube used in the experiment was a 15.88 mm diameter copper tube made of phosphorus deoxidized copper (C1220), commonly used in plate heat exchangers. The STS tube, made of austenitic stainless-steel material (STS304), was designed to accommodate the insertion of the copper tube at the top. A total of 100 sets of STS tubes and copper tubes had been prepared. For the brazing joint, a 1 mm thick spring-type copper with 20% Ag as the filler metal was used. The filler metal for the brazing joint was 1 mm thick spring-type copper containing 20% Ag, SK Brazing's H-type, high-temperature flux was applied to the joint area for oxidation prevention and reduced cooling speed.

The brazing joint experiment of copper–stainless-steel tubes using high-frequency induction heating was conducted after impurities were removed. The experiment aimed to secure training data for both normal and defective joints by varying the parameters of the brazing experiment in order to induce various temperature changes at the joint. The key parameters selected for the experiment were the brazing time (8~15 s) and the center height of the high-frequency electrode (−6~6 mm), which would have an influence on the temperature changes at the joint. The experiment was carried out by altering these parameters according to the experimental plan. Using a full factorial-design method, we conducted five repeated experiments with variations in current, voltage, the center height of the high-frequency electrode, and brazing time. The remaining samples were adjusted by modifying parameters to match the quantities of results under defective and satisfactory brazing conditions. In Figure 2a, a schematic diagram represents the brazing joint of copper–stainless-steel tubes using high-frequency induction heating in a plate heat exchanger. Figure 2b provides a schematic description of the center height of the high-frequency electrode, used as a parameter. Table 1 lists the parameters used in the high-frequency induction-heating brazing experiment, and Table 2 presents the chemical composition of the copper tube, stainless-steel tube, and filler metal used in the experiment.

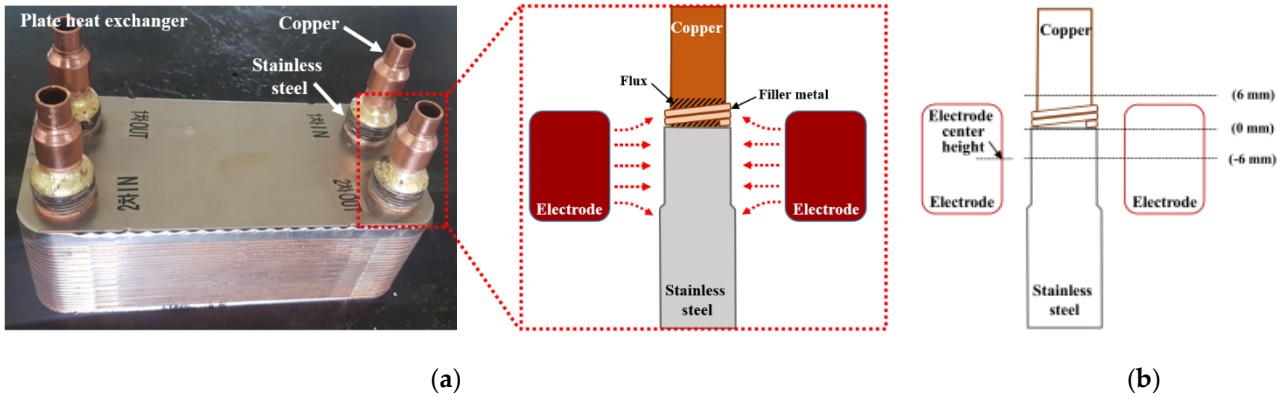

(**a**)                                                                                           (**b**)

**Figure 2.** Schematic diagram of high-frequency induction-heating brazing experiment: (**a**) High-frequency induction-heating brazing; (**b**) Electrode center height.

**Table 1.** Process parameters for brazing experiments.

| Variable | Level |
|---|---|
| Current(A) | 32 |
| Voltage(V) | 259 |
| Brazing time(sec) | 8–15 |
| Fixed variable | Filler metal: Spring-type<br>Filler metal point: 0 mm<br>Electrode center point: 0 mm<br>Outside temperature: 27.2 °C<br>Humidity 68% |

**Table 2.** Chemical composition.

| Material | Cu | P | Cr | Ni | Si | Mn | N | C | S | Ag | Zn | In | Sn | Fe |
|---|---|---|---|---|---|---|---|---|---|---|---|---|---|---|
| Copper tube | Bal. | 0.04 | | | | | | | | | | | | |
| Filler metal | Bal. | | | | | | | | | 20 | 35 | 0.5 | 0.5 | |
| STS tube | | 0.045 | 18.0 | 8.0 | 7.5 | 2.0 | 0.1 | 0.08 | 0.03 | | | | | Bal. |

### 2.2. Joint Defect Discrimination Method

Through high-frequency induction heating, brazing joints were performed on copper–stainless-steel tubes, and the penetration depth of the joints was measured using acquired infrared thermal imaging data to develop a CNN model for detecting joint defects. To measure the penetration depth of the filler metal in the copper–stainless-steel tube joints performed according to parameters, axial cutting was performed using a high-speed cutter (Allied (Compton, CA, USA), Techcut5). The cut surface was polished (R&B (Daejeon, Republic of Korea), RB 204 METPOL-II) to the extent where the penetration depth could be confirmed. The measurement of penetration depth was carried out using a digital optical microscope (Leetech, portable welding microscope) with a resolution of 2 megapixels, without etching the specimen. In this study, the joint between the copper tube and stainless-steel tube was designed with a length of 10 mm, and according to the regulations of ISO 18279, a joint was considered satisfactory if the depth of penetration overlapped more than 70%. Therefore, joint-experiment results with a filler metal penetration depth of 7 mm or more, without apparent issues, were considered satisfactory joints. Problems that could occur in welding and joining areas generally included the influx of external impurities, bubbles, and cracks, which could act as factors reducing the strength of the joint. However, in the brazing joint of the copper–stainless-steel tube conducted in this study, even if external impurities and bubbles occurred, as long as the filler metal showed

a penetration depth of 70% or more, it indicated sufficient joint strength, as suggested by ISO 18279. Therefore, in this study, the judgment of joint defects was based on whether the penetration depth of the filler metal was 70% or more and on the integrity of the internal and external shape. Figure 3 illustrates the locations for measuring the penetration depth of brazing joints.

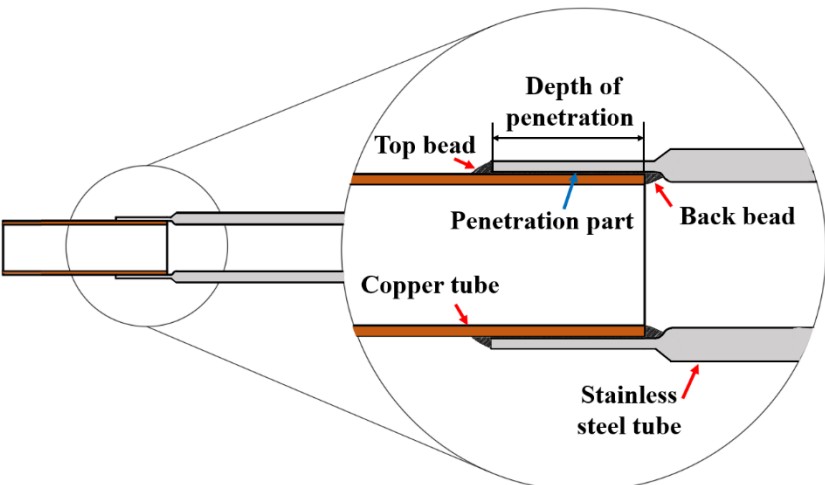

**Figure 3.** Method for measuring cross-sectional shape and penetration depth of brazing joints.

## 3. CNN Model Design

### 3.1. Infrared Thermal-Imaging-Data Collection

In order to detect defects in the brazing joints of copper–stainless-steel tubes using an infrared thermal imaging camera, this study collected thermal imaging data during the brazing process based on the selected parameters. The collection of thermal imaging data commenced as the high-frequency induction-heating brazing-joint-experiment began, with the jig fixed in place according to the selected electrode-center-height parameter. The thermal imaging data were saved as 80 × 80-pixel images, with a frame rate of 10 frames per second. To utilize the temperature changes during the cooling process after the joint had been made, the thermal imaging data were collected for an additional 3 s (30 frames) after the end of the brazing joint time. A total of 100 high-frequency induction-heating brazing experiments were conducted for the collection of thermal imaging data on copper–stainless-steel tube brazing joints, resulting in 47 sets of normal joint data and 53 sets of defective joint data. Figure 4 illustrates the results of the brazing joints in copper–stainless-steel tubes. In this study, the presence of joint defects was determined based on the filler metal's penetration depth in the completed copper–stainless-steel tube joints, and the thermal imaging data results were presented accordingly. Figure 5 depicts the thermal imaging data collected during the process of brazing joints of copper–stainless-steel tubes.

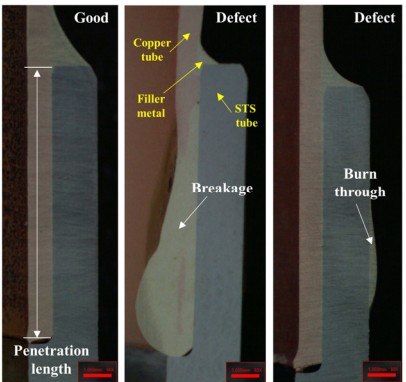

**Figure 4.** Brazing joint results.

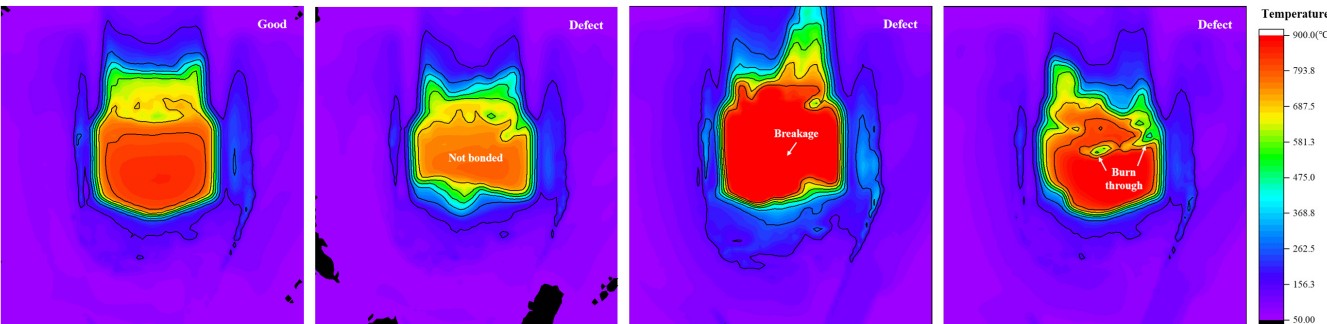

**Figure 5.** Thermal image results of brazing joints.

*3.2. Infrared Thermal Imaging Data Preprocessing*

In general, image data fed into CNN models were represented in RGB-color format, consisting of three channels. However, the infrared thermal imaging data used in this study were employed as single-channel images. Over the course of 100 high-frequency induction-heating brazing experiments, 10,233 frames of 80 × 80-pixel thermal imaging data were acquired. Each frame of thermal imaging data represented a unique temperature change, and thus, the labeling was performed at a rate of 1 label per frame. The labeling of the thermal imaging data was evaluated based on the joint-defect-discrimination method presented in Section 2.2. The normal joint thermal imaging data were classified as 0, and the defective joint thermal imaging data were classified as 1. Before feeding the thermal imaging data into the convolutional layers, data-labeling preprocessing was carried out. As a result, a total of 4796 sets of normal joint data and 5437 sets of defective joint data were obtained. Figure 6 illustrates the preprocessing method for the collected infrared thermal imaging data from high-frequency induction-heating brazing experiments.

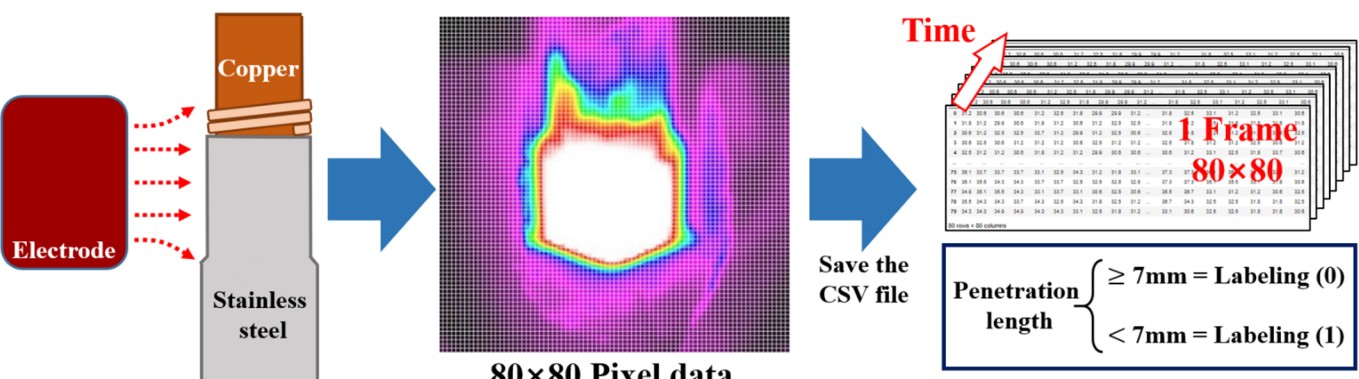

**Figure 6.** Storage structure of thermal image data.

A total of 10,233 thermal imaging data, after labeling preprocessing, were categorized into training data, validation data, and test data, for the development and performance evaluation of the brazing-joint-defect prediction model. The data-categorization process ensured that the normal-joint data and the defective-joint data were not biased, and no human intervention was involved, except for this aspect. The training data and test data were divided into an 8:2 ratio of the entire dataset, and within the separated training data, the training data and validation data were further divided into an 8:2 ratio. As a result, 10,233 thermal imaging data were categorized into 6548 training data, 1638 validation data, and 2047 test data.

*3.3. Data Normalization*

The infrared thermal-imaging data collected from the copper–stainless-steel tube brazing joint experiments using high-frequency induction heating were captured in an

$80 \times 80$-pixel format. This meant that in addition to the joint area, it also included temperature information from the surrounding environment and the high-frequency electrodes, which were unnecessary for the prediction model and could potentially slow down the training process by widening the temperature range of the data within the $80 \times 80$-pixel area. In this study, we applied the max-normalization method to normalize the thermal imaging data to a range between 0 and 1. Data normalization was performed based on the highest recorded temperature among the 10,233 frames collected during the experiments. The normalized data were then reshaped into a one-channel dimension before being used as training data for developing the CNN model.

*3.4. CNN Model Concept Design*

The convolutional layer used in this study included a convolution kernel, and the principle that the kernel would be affected by the weight coefficient and bias is expressed by Equation (1) [24].

$$
\begin{aligned}
\mathrm{F}^{l+1}(i,j) &= \left[ \mathrm{F}^l \otimes w^{l+1} \right](i,j) + b \\
\sum_{n=1}^{n^1} \sum_{x=1}^{k} \sum_{y=1}^{k} & \left[ F_n^1(s_1 \times i + x, s_1 \times j + y) w_n^{l+1}(x,y) \right] + b
\end{aligned}
\tag{1}
$$

where $\mathrm{F}^l$ is the input image of the convolutional layer of the $l+1$ layer, $\mathrm{F}^{l+1}$ is the output image of the convolutional layer of the $l+1$ layer, and $\otimes$ is the convolutional operation. $\mathrm{F}(i,j)$ represents the pixel of the corresponding feature map, $w^{l+1}$ is the convolution kernel weight coefficient of the $l+1$ layer, and $b$ is the bias vector. Additionally, $k$ is the convolution kernel size, $s_1$ is the convolution stride, and $n^1$ is the number of convolution kernels [25]. When the convolutional operation was complete, $F(i,j)$ underwent an activation function to obtain $A(i,j)$. In this study, the ReLU function was used as the activation function and is represented by Equation (2).

$$
A(i,j) = ReLU(F(i,j)) = \max(0, F(i,j))
\tag{2}
$$

After the convolutional layer, a pooling layer was applied, which reduced the complexity of the model by removing images of relatively low importance. In this study, the feature was extracted using max-pooling and is represented by Equation (3).

$$
P^l(i,j) = \left[ \sum_{x=1}^{d} \sum_{y=1}^{d} A^l(s_2 \times i + x, s_2 \times j + y)^m \right]^{\frac{1}{m}}
\tag{3}
$$

where $P^l(i,j)$ is the output feature map of the $l^{th}$ pooling layer, $d$ is the pooling window size, $s_2$ is the translation stride of the pooling window, and $m$ converges to $\infty$. In this study, the data that has passed through the convolution layer and pooling layer is utilized in the Fully Connected (FC) layer. The CNN-based prediction model for brazing joint defects using thermal imaging data involved $80 \times 80$-pixel thermal images. The CNN-based defect-detection and -prediction model for brazing joints using thermal imaging data employed $80 \times 80$-pixel thermal images and utilized two convolutional layers. It used a $3 \times 3$ kernel size for convolution, which was known for its optimal functional efficiency [9]. The same padding was used to maintain image size, and the ReLU function was used as the activation function. The resulting images were down-sampled using $2 \times 2$ max-pooling and passed to the FC layer, which consisted of three layers in total after being flattened. Dropout was introduced between layers to prevent overfitting during the image-transfer process. The activation function used in the 1st and 2nd FC layers was ReLU, and the Adam (adaptive moment assessment) optimizer was employed to minimize the loss function and update the model weights. Finally, the 3rd FC layer utilized a sigmoid function to predict the occurrence of defects in the joint, classifying them as 0 or 1. Figure 7 represents a schematic of the CNN model developed to predict the occurrence of defects in a brazing joint using

thermal imaging data obtained from the copper–stainless-steel tube brazing experiments conducted with high-frequency induction heating.

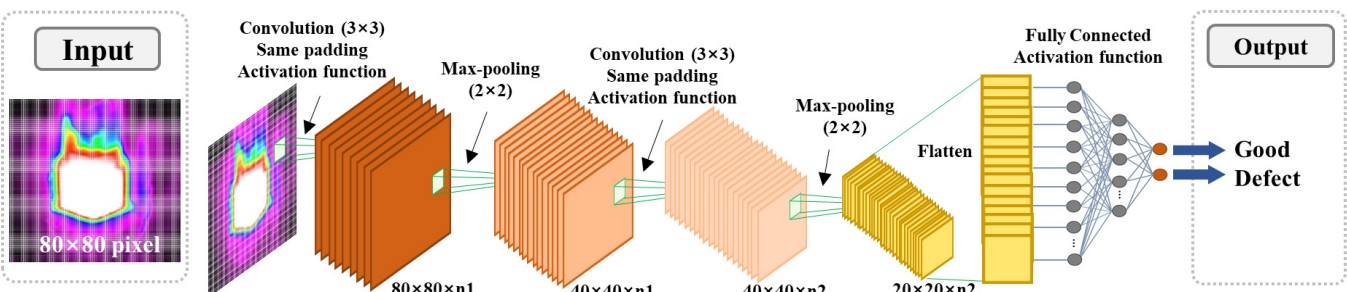

**Figure 7.** Schematic diagram of CNN model.

### 3.5. CNN Model Training

In order to select the hyperparameters for a CNN-based prediction model designed to discriminate brazing joint defects in copper–stainless-steel tubes, the model was trained, and the prediction results were compared. For the purpose of hyperparameter selection, the first convolutional layer and the second fully connected layer were maintained with fixed structures. A sigmoid function was used as the activation function for discerning the presence of defects in the joints. The hyperparameters were chosen based on the number of kernel filters in the second convolutional layer and the number of nodes in the first fully connected layer. To observe clear differences in the results based on the hyperparameters, the values with a two-fold difference were selected. Six different CNN-based prediction models with varying structures were compared using the same training, test, and validation data, all obtained during the data-preprocessing stage. The number of epochs was set to 100 across all models. Table 3 illustrates the structures of these six different CNN models. The computer configuration used for training the CNN-based prediction model for brazing joint defects was as follows: Python programming language, Windows 10 OS, 11th Gen Intel (R) Core (TM) i7-1165G7 2.80 GHz 2.80 GHz, 16.0 GB RAM.

**Table 3.** CNN model structure for selection of hyperparameters.

| CNN Model | Conv. 1 Layer (Kernel Filter) | Conv. 2 Layer (Kernel Filter) | FC 1 Layer (Node) | FC 2 Layer (Node) | Activation Function |
|---|---|---|---|---|---|
| model-1 | 64 | 32 | 256 | 128 | Sigmoid |
| model-2 | 64 | 32 | 512 | 128 | Sigmoid |
| model-3 | 64 | 32 | 1024 | 128 | Sigmoid |
| model-4 | 64 | 64 | 256 | 128 | Sigmoid |
| model-5 | 64 | 64 | 512 | 128 | Sigmoid |
| model-6 | 64 | 64 | 1024 | 128 | Sigmoid |

## 4. Brazing Joint Defect-Discrimination-Model Prediction Results

### 4.1. Performance of CNN Models

The performances of copper–stainless-steel tube brazing-joint-defect prediction models, designed with different hyperparameter structures, were compared. To evaluate the performances of these prediction models, we used validation data and test data, which had been pre-allocated to avoid issues, such as overfitting, when the models were re-evaluated on the training data. In Figure 8, the results of these prediction models are presented, demonstrating the utilization of the validation data and test data for the models with the selected hyperparameters. The performance comparison based on accuracy revealed that there was little difference in the accuracy achieved using the validation data versus the test data. This suggested that all six prediction models were appropriately designed. When using test data, CNN model-3 exhibited the highest accuracy performance, reaching 99.12%, while CNN model-5 showed the lowest accuracy performance at 98.53%.

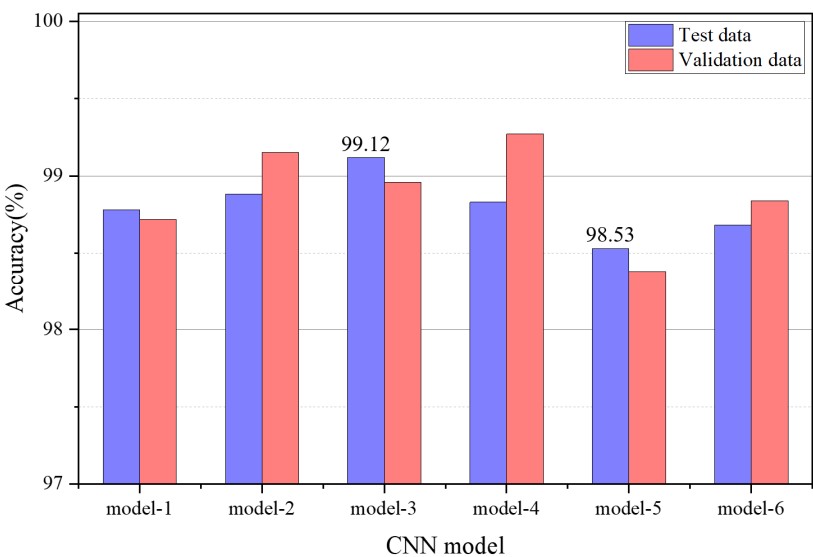

**Figure 8.** Accuracy performance of CNN models with different structures.

*4.2. Selection of CNN Model Structure*

To compare the predictive performance of the six different CNN models with varying architectures, an analysis was conducted using f1_score and recall to ensure that the result data were not biased. The test data predictions from the six CNN models yielded results consistent with the earlier performance evaluation. CNN model-3 exhibited the highest f1_score at 0.9918, while CNN model-5 had the lowest f1_score at 0.9864. However, in terms of the recall for predicting faulty joints, CNN model-5 demonstrated the best performance at 99.82%, whereas CNN model-4 had the lowest recall at 98.9%. In typical predictive models, a superior model has often been selected based on the excellence of the f1_score when assessing predictive performance and ensuring unbiased data representation. However, in this study, the critical concern was not only predicting the performance of the brazing joints but also avoiding the misclassification of actual faulty results as normal joints. Hence, recall performance became the most crucial factor. The recall of the CNN models represented the proportion of correctly predicted faults among the actual faulty conditions. Particularly in situations where misjudging actual faulty data as normal could lead to substantial damage, the importance of recall would be heightened. In industrial settings, predicting actual normal joint results as faulty would be less problematic than incorrectly classifying actual faulty results as normal joints, which could lead to significant product failure and defects. Considering the prediction task of identifying faults in the brazing joints of tubing conducted in this study, misclassifying actual faulty results as normal was a more significant issue than predicting normal joints as faulty. Therefore, the importance of recall was paramount. Despite CNN model-3 exhibiting the best overall predictive performance and f1_score, CNN model-5, with its relatively lower predictive performance but superior recall, aligned better with the goals of this study. As a result, CNN model-5 was selected as the final model structure. Figure 9 illustrates the f1_score and recall performances of different CNN model structures using the test data.

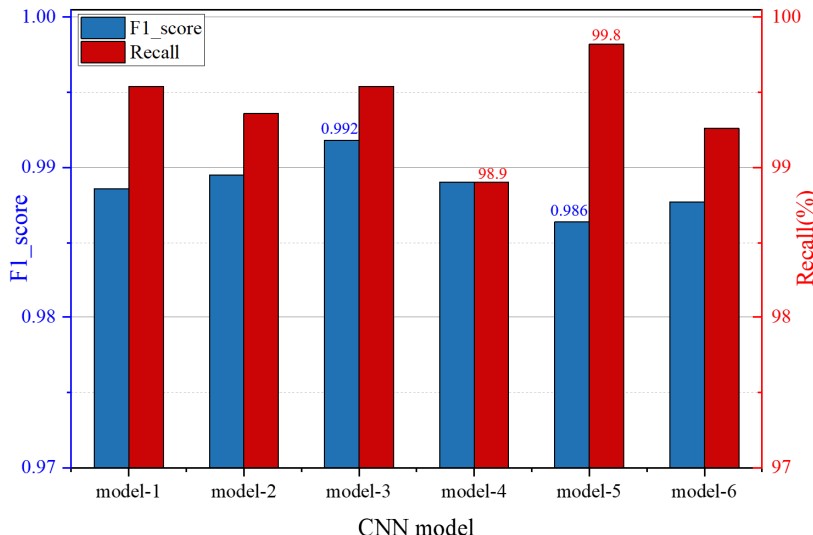

**Figure 9.** Prediction performance on test data of CNN models with different structures.

To analyze the research results presented in Figure 9, we conducted a confusion matrix analysis. A confusion matrix analysis has commonly been used to help evaluate a model's performance by comparing the model's predictions with the actual result data and can help identify any bias in the data analysis. In this study, since a sigmoid function was used as the activation function for discriminating the presence of defects in the joints, a $2 \times 2$ matrix of the confusion matrix results was used. All six structurally different predictive models did not show any bias in the use of normal joint data and defective joint data, confirming the absence of bias in the data analysis. Furthermore, as revealed by the results of the confusion matrix analysis, the sum of the FNs (false negatives) and FPs (false positives), which represented the model's accuracy performance, was the lowest for CNN model-3. However, when considering the cases in which defective joints had been incorrectly predicted as normal joints (FN), CNN model-5's structure showed even lower values than CNN model-3's structure, making the recall performance superior in CNN model-5. Figure 10 illustrates the results of the confusion matrix analysis for the CNN prediction models.

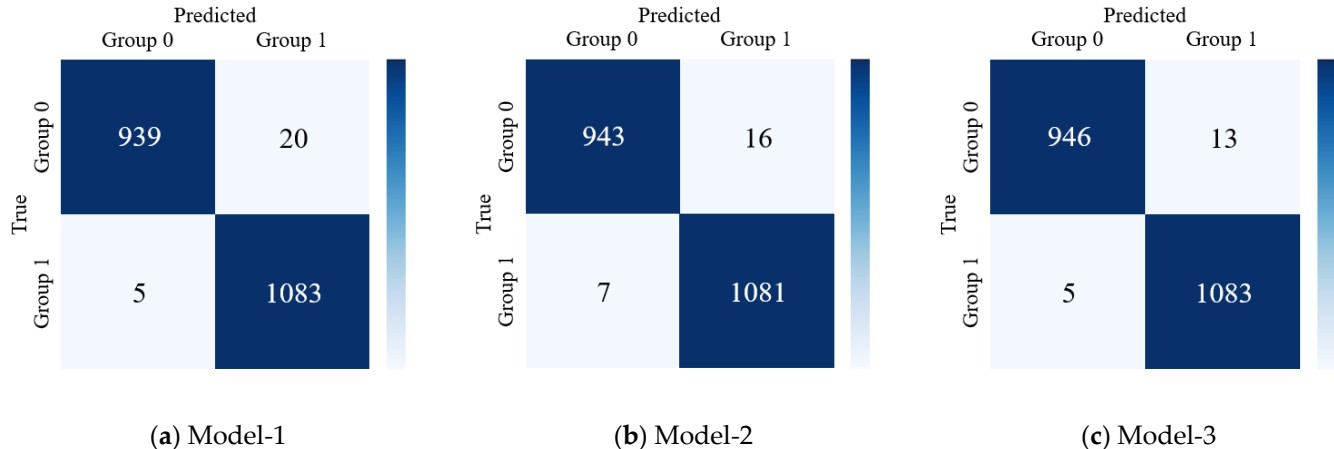

(**a**) Model-1  (**b**) Model-2  (**c**) Model-3

**Figure 10.** *Cont.*

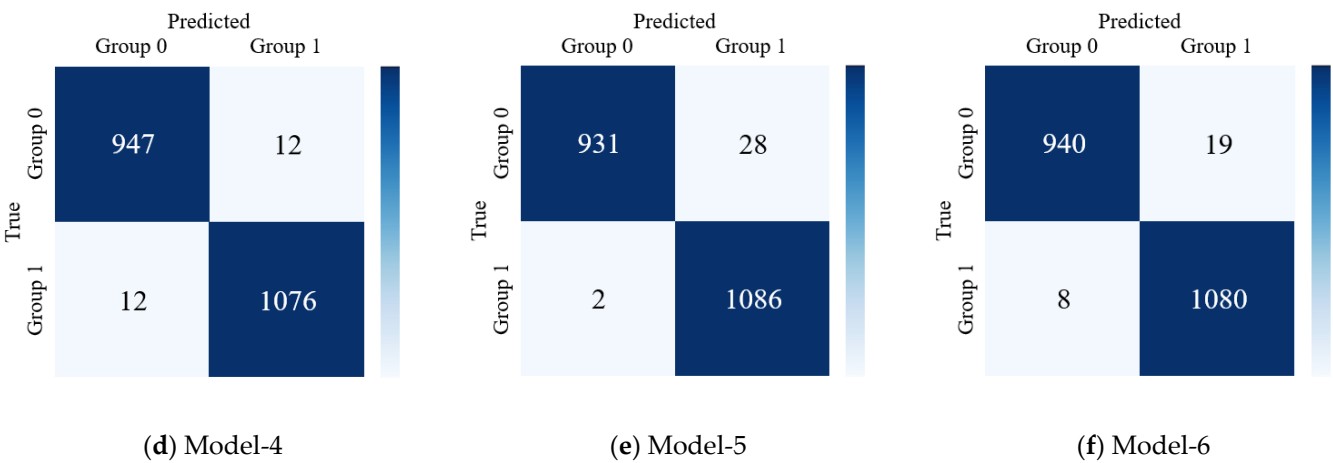

(**d**) Model-4          (**e**) Model-5          (**f**) Model-6

**Figure 10.** Results of confusion matrix analysis.

In this study, CNN model-5 was selected as the final CNN model structure for predicting defects in copper–stainless-steel tube brazing joints using infrared thermal imaging data. CNN model-3's structure used 64 kernel filters in the first convolutional layer, and 64 kernel filters in the second convolutional layer. In the fully connected layer, the first layer had 512 nodes, the second layer had 128 nodes, and finally, the model used a sigmoid function as the activation function to predict the occurrence of defects in copper–stainless-steel tube brazing joints utilizing high-frequency induction heating.

In Figure 11, the performance and losses of CNN model-5 as a function of epochs using test data are presented. It was observed that as the number of epochs and accuracy increased, the losses decreased, indicating that overfitting had not occurred during the training process using the training data. Furthermore, these results demonstrated that a CNN-based predictive model using the infrared thermographic data was capable of predicting defects in the joints.

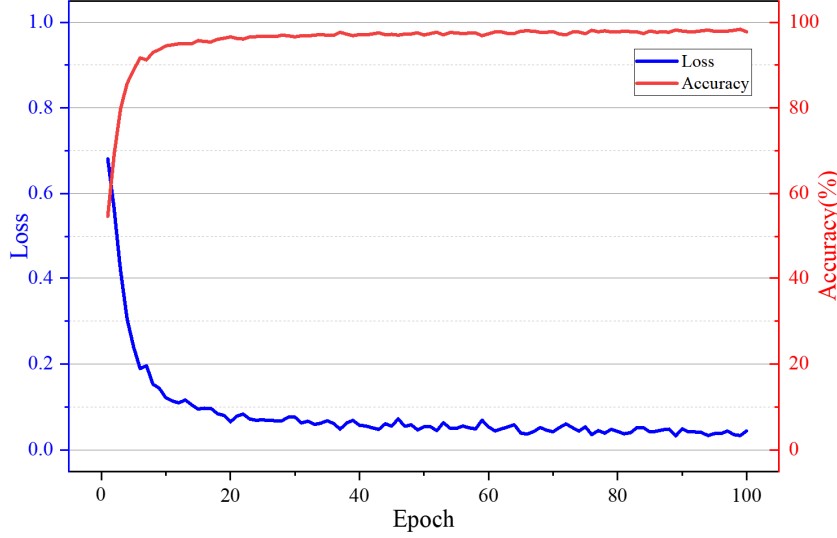

**Figure 11.** Prediction performance of CNN models that improve with increasing epoch.

Figure 12 illustrates the average defect-detection-time performance of the different CNN models. The results comparing the detection-time performance of joint defects using the test data indicated that model-1 had exhibited the fastest detection time. While there was a tendency for the defect-detection time to increase with an increase in the number of kernel filters in the selected convolutional layer and the number of nodes in the second fully connected layer, as hyperparameters, the differences were deemed very marginal.

All six different CNN models showed defect-detection-time performances in the range of the 2 s mark. Based on these results, it was inferred that the CNN models for defect discrimination could be used in real-time applications.

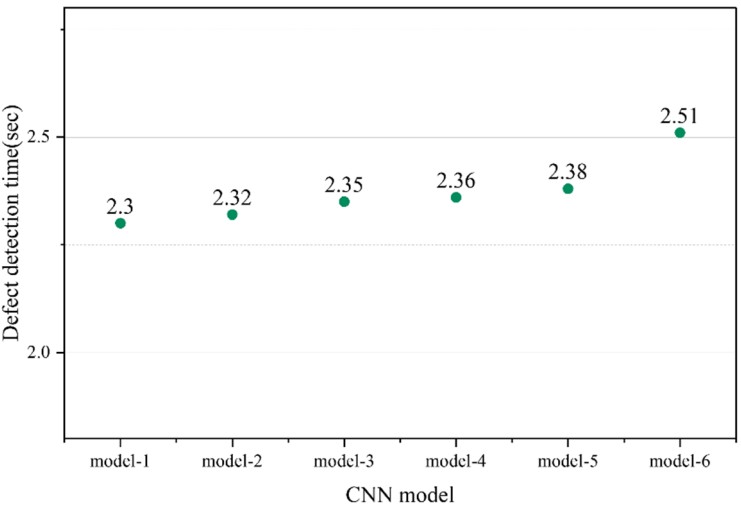

**Figure 12.** Defect detection time performance of CNN models with different structures.

## 5. Conclusions

In this study, we proposed a system for detecting defects in the brazing joints of copper and stainless-steel tubes using high-frequency induction heating. We developed a defect prediction model based on CNNs using the collected infrared thermographic data for discerning defects in copper–stainless-steel tube brazing joints. The results of the validation of the prediction performance are summarized as follows:

We obtained 4796 normal joint data and 5437 defective joint data in the form of $80 \times 80$-pixel infrared thermographic data during 100 high-frequency induction-heating brazing experiments. A total of 10,233 infrared thermographic data were classified into 6548 training data, 1638 validation data, and 2047 test data, for model development.

The CNN-based predictive model for brazing-joint defects selected six combinations of hyperparameters, including the number of kernel filters in the convolutional layer and the number of nodes in the fully connected layer, and compared their predictive performance. Although there were differences in the accuracy performance depending on the model, an overall excellent accuracy performance was observed, ranging from a minimum of 98.53% to a maximum of 99.12%.

Predicting defects in the brazing joints of copper tubes and stainless-steel tubes prioritized recall performance over accuracy since predicting the actual defects as normal joints could lead to significant issues in the final product. The final selected model demonstrated the most outstanding recall performance, reaching 99.82% for brazing joint defects.

As the number of kernel filters in the selected convolutional layer and the number of nodes in the second fully connected layer increased as hyperparameters, there was an increase in defect detection time. However, all six different CNN models exhibited defect-detection-time performance in the range of 2 s, indicating the capability for real-time prediction of joint defects.

Through a study on the prediction of defects in brazing joints using a CNN-based approach with infrared cameras, this research demonstrates a method and its feasibility for discerning hard-to-observe joint defects in tube structures. The proposed method provides a novel idea for improving the efficiency of defect detection in industrial sectors where welding and joining processes are performed, offering the potential for enhanced productivity in identifying defects that are challenging to confirm visually.

**Author Contributions:** Conceptualization, J.K.; experiment, C.-W.L.; software, S.W.; validation, J.K. and C.-W.L.; paper research, S.W., J.K. and C.-W.L.; Data analysis, C.-W.L.; writing—original draft preparation, C.-W.L.; writing—review and editing, C.-W.L. and S.W.; supervision, J.K.; project administration, J.K. All authors have read and agreed to the published version of the manuscript.

**Funding:** This study has been conducted with the support of the Korea Institute of Industrial Technology as "Autonomous Manufacturing Technology based on DNA Platform (kitech EH-230006)".

**Data Availability Statement:** Data are contained within the article.

**Conflicts of Interest:** The authors declare no conflict of interest.

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
