# Peer review of "A Study on Defect Detection of Dissimilar Joints in Cu-STS Tubes Using Infrared Thermal Imaging of Induction Heating Brazing"

_processes, doi:10.3390/pr12010163_

Round 1

Reviewer 1 Report

Comments and Suggestions for Authors

This work uses the Convolutional Neural Network (CNN) model to detect defects that can occur during the brazing process of copper and stainless steel tubes. The content presented by the authors is well described and the article is easy to understand. I therefore recommend that the article be published and that the authors carry out a basic revision of the english writing to improve the fluidity of the text.

Comments on the Quality of English Language

I suggest a minor revision of the text of the article.

Author Response

We sincerely appreciate your thoughtful and meticulous review of our submitted paper titled "A Study on Defect Detection of Dissimilar Joints in Cu-STS Tubes Using infrared thermal imaging of induction heating brazing." We have carefully addressed the points raised and made the necessary revisions and improvements. We kindly request your reconsideration and would like to express our sincere gratitude for your hard work.

Reviewer 2 Report

Comments and Suggestions for Authors

This study reports on the defect detection that has always been concerned in brazing. The research is very attractive, but there is still a small amount of work that needs to be modified before acceptance.

(1)  As for Fig. 4, rules should be added on the figures.

(2)    In this work, authors develop six CNN models to verify its accuracy. It will be better if a specific case can be provided to confirm the accuracy of the CNN model, especially model-5.

(3)    It is recommended to provide more thermal images representing various brazing results, such as good and defects cases in fig. 4.

Author Response

(The authors gave the same response as above.)

Reviewer 3 Report

Comments and Suggestions for Authors

The article discusses a detection method based on a convolutional neural network model.

The work is devoted to a current topic, but before accepting the article it needs to be finalized.

1. The abstract is quite short and does not fully reflect the results obtained. It needs to be adjusted and expanded a little. Describe the results obtained more fully in the abstract.

2. In the introduction, you write about using machine learning to predict defects. It is necessary to describe in more detail what and on the basis of what forecasting is carried out. That is, tomography results are used as initial data and, based on them, defects are predicted throughout the entire volume of the weld? Or is the forecast based on welding modes? Describe this in more detail in the introduction. You can also cite more modern publications concerning the reliability of materials and technologies based on them. This will show how important it is to understand defects and how they affect properties. For example https://doi.org/10.3390/math11153317;  https://doi.org/10.1016/B978-0-12-820963-9.09993-4;

3. It would be nice to expand the introduction a little and provide more information from similar works. Write down what accuracy the forecasting provided, how long the process took, and what benefits it allowed to obtain.

4. Describe the Methods and Materials section in more detail. Provide all the equipment used, check that you have indicated all its operating modes. For the equipment used, indicate the country of origin and manufacturer in parentheses. Describe what machine learning methods you used. The general plan of experimental work will be given at the very beginning of this section. Indicate how many samples you used for each type of study.

5. In Figure 4, show the size guide.

6. It would be good to show the defects under study at high magnification, since in Figure 4 the defects are practically invisible.

7. It would be worthwhile to provide more comparative data on the processing speed and accuracy of the methods used in the conclusions.

8. Also, the article mainly describes simply welding and soldering defects. But at the same time, defects can be different and difficult to identify. These could be foreign inclusions, gas bubbles, or cracks. At the same time, small cracks are quite difficult to detect, but they make a significant contribution to reducing the strength of the joint. This issue is practically not addressed in the article. A consideration of this issue should be added. Describe the types of defects you detect, describe which defects your methods can predict and with what accuracy.

Author Response

(The authors gave the same response as above.)

Round 2

Reviewer 3 Report

Comments and Suggestions for Authors

The authors revised the article and made the necessary changes. The article may be published in my opinion.